# Population Genetics and Anastomosis Group’s Geographical Distribution of *Rhizoctonia solani* Associated with Soybean

**DOI:** 10.3390/genes13122417

**Published:** 2022-12-19

**Authors:** Aqleem Abbas, Xiangling Fang, Shehzad Iqbal, Syed Atif Hasan Naqvi, Yasir Mehmood, Muhammad Junaid Rao, Zeshan Hassan, Roberto Miño Ortiz, Alaa Baazeem, Mahmoud Moustafa, Sulaiman Alrumman, Sally Negm

**Affiliations:** 1State Key Laboratory of Grassland Agroecosystems, Key Laboratory of Grassland Livestock Industry Innovation, Ministry of Agriculture and Rural Affairs, College of Pastoral Agriculture Science and Technology, Lanzhou University, Lanzhou 730000, China; 2College of Plant Sciences and Technology, Huazhong Agricultural University, Wuhan 430070, China; 3Department of Plant Pathology, Faculty of Agricultural Sciences and Technology, Bahauddin Zakariya University, Main Campus, Bosan Road, Multan 60800, Pakistan; 4State Key Laboratory for Conservation and Utilization of Subtropical Agro-Bioresources, Guangxi Key Laboratory of Sugarcane Biology, College of Agriculture, Guangxi University, Nanning 530021, China; 5College of Agriculture, Bahauddin Zakariya University, Multan, Bahadur Sub Campus, Layyah 31200, Pakistan; 6Instituto de Ciencias Biológicas, Universidad de Talca, Talca 3460000, Chile; 7Department of Biology, College of Science, Taif University, Taif 21944, Saudi Arabia; 8Department of Biology, Faculty of Science, King Khalid University, Abha 62529, Saudi Arabia; 9Department of Botany and Microbiology, Faculty of Science, South Valley University, Qena 83523, Egypt; 10Department of Life Sciences, College of Science and Art Mahyel Aseer, King Khalid University, Abha 62529, Saudi Arabia; 11Unit of Food Bacteriology, Central Laboratory of Food Hygiene, Ministry of Health, Branch in Zagazig, Sharkia 44511, Egypt

**Keywords:** soybean, *R. solani*, anastomosis groups, distribution, genetic diversity, phylogeny

## Abstract

*Rhizoctonia solani* is a species complex composed of many genetically diverse anastomosis groups (AG) and their subgroups. It causes economically important diseases of soybean worldwide. However, the global genetic diversity and distribution of *R. solani* AG associated with soybean are unknown to date. In this study, the global genetic diversity and distribution of AG associated with soybean were investigated based on rDNA-ITS sequences deposited in GenBank and published literature. The most prevalent AG, was AG-1 (40%), followed by AG-2 (19.13%), AG-4 (11.30%), AG-7 (10.43%), AG-11 (8.70%), AG-3 (5.22%) and AG-5 (3.48%). Most of the AG were reported from the USA and Brazil. Sequence analysis of internal transcribed spacers of ribosomal DNA separated AG associated with soybean into two distinct clades. Clade I corresponded to distinct subclades containing AG-2, AG-3, AG-5, AG-7 and AG-11. Clade II corresponded to subclades of AG-1 subgroups. Furthermore, AG and/or AG subgroups were in close proximity without corresponding to their geographical origin. Moreover, AG or AG subgroups within clade or subclades shared higher percentages of sequence similarities. The principal coordinate analysis also supported the phylogenetic and genetic diversity analyses. In conclusion, AG-1, AG-2, and AG-4 were the most prevalent AG in soybean. The clade or subclades corresponded to AG or AG subgroups and did not correspond to the AG’s geographical origin. The information on global genetic diversity and distribution will be helpful if novel management measures are to be developed against soybean diseases caused by *R. solani*.

## 1. Introduction

Soybean (*Glycine max* L.) is one of the world’s most significant oilseed crops, accounting for 25% of all edible oil production [1]. About 176.6 million tons of soybeans are produced over 75.5 million hectares of fertile land each year [2,3]. *R solani* Kuhn [teleomorph, *Thanatephorus cucumeris* (Frank) Donk] is a serious threat to soybean production worldwide. The fungus causes blights (foliar and web), pre-and post-emerging damping-off, root and hypocotyl rot diseases of soybean [4,5]. These diseases caused massive yield losses in soybeans all over the world. For example, in Brazil and the United States, only foliar blight has resulted in 30 to 69% yield losses [6,7,8,9]. Moreover, these diseases are difficult to control because of soil born nature of *R. solani* and the broad host range [4]. Fungicides have been widely used to manage these diseases [10,11]. However, fungicides have caused severe environmental and health concerns. The most cost-effective and environmentally sustainable option to manage *R. solani* is breeding resistant cultivars [12]. However, understanding the genetic diversity of *R. solani* is critical if novel management measures, such as developing *Rhizoctonia* -resistant cultivars, are to be developed. *R. solani* exhibits tremendous genetic diversity and is classified into different anastomosis groups (AG). To date, 14 anastomosis groups (AG 1 to 13 and AG-BI) have been identified based on the fusion of hyphae, morphology, virulence (pathogenicity), physiology, and DNA homology [13,14]. Some of the AG have been further divided into subgroups based on anastomosis frequency, physiological and morphological features, pathogenic, bimolecular, biochemical, genetic, and DNA homology characteristics [15,16].For example, AG-1 has been divided into six subgroups: IA, IB, IC, ID, IE, and IF [17]. Similarly, AG-4 has been divided into three subgroups: HGI, HGII, and HGIII [18], and AG-2 has been divided into nine subgroups such as 1, 2, t, Nt, 2IIIB, 2IV, 2LP, 3 and 4 [18]. AG-2, AG-4, AG-5, AG-3, AG-7, and AG-11 are causing damping-off and root, and hypocotyls rot, whereas AG-1 is responsible for the foliar and web blight of soybean [5,19,20,21,22].

To evaluate genetic diversity and to characterize AG of *R. solani*, numerous molecular markers were used, which included inter-simple sequence repeats (ISSR) [23], simple sequence repeats (SSR) [24,25], single nucleotide polymorphisms (SNPs) [26], amplified fragment length polymorphism (AFLP) [27], restriction fragment length polymorphisms (RFLP) [28], randomly amplified polymorphic DNA (RAPD) markers [29], electrophoretic karyotype [30], DNA-DNA hybridization [31] and sequence analysis of rDNA ITS1-5.8 S-ITS2region [32,33,34]. Among these molecular markers, sequence analysis of rDNA ITS1-5.8 S-ITS2 region, because of their presence of multiple copies of tandem repeats within the genome of all fungi, has proven to be a more powerful tool for genetic diversity and phylogenetic studies of AG and AG subgroups of *R. solani* [15,18,34,35,36,37]. Moreover, it validates the grouping of AG on the basis of classical hyphal anastomosis reactions [38,39]. Furthermore, the rDNA ITS1-5.8S-ITS2 region sequences evolve quickly and are bordered by highly conserved nucleotide sequences [40]. They are found between the 18S and 5.8S rRNA genes (ITS1) and the 5.8S and 28S rRNA genes (ITS2) [38,41].

Information on the genetic diversity and distribution of *R. solani* AG associated with soybean of a particular country is available [1,42,43,44]. However, there was no attempt to investigate the global genetic diversity and distribution of *R. solani* AG associated with soybean. Considering the genetic diversity and different diseases causing abilities of *R. solani* AG on soybean mentioned above, the current study was aimed (1) To determine most frequently reported and dominant AG associated with soybean; (2) To explore the genetic diversity of AG based on rDNA ITS1-5.8S-ITS2 sequence analysis; (3) To determine the relationship between geographical origin and genetic diversity of AG. 

## 2. Materials and Methods

### 2.1. Data Collection

Soybean (*G. max* L.) infected by *R. solani* was targeted in this study. Relevant literature was searched in Google Scholar (http://scholar.google.com; accessed on 11 November 2021), Web of Science (http://apps.webofknowledge.com; accessed on 12 November 2021), China National Knowledge Infrastructure (CNKI) (https://www.cnki.net; accessed on 12 November 2021), Scholarly, Academic Information Navigator (CiNii) (http://ci.nii.ac.jp/en; accessed on 13 November 2021), PubMed (https://pubmed.ncbi.nlm.nih.gov/; accessed on 14 November 2021), and Scopus (https://www.elsevier.com/en-gb/solutions/scopus; accessed on 15 November 2021) using the main keywords “soybean-*R. solani*, ” and the “soybean-anastomosis groups (AG)” solely and in combination. The search for literature was limited from January 2001 to October 2021. To be included in this study, the literature had to meet the following criteria: (i) Only articles published in peer-reviewed journals were chosen; (ii) Articles mentioning the accession numbers of AG and the sequencing data of AG was publicly available in GenBank; (iii) Articles mentioning the geographical origin, isolates and AG that could cause symptoms on soybean; (iv) Articles mentioning about pathogen isolation from the soil (e.g., rhizosphere soil, topsoil), root, and shoot of the symptomatic soybean plants. The articles from the databases mentioned above were imported into the EndNote X9 software to acquire information of AG, isolate, geographical origin, and isolation sources. The information on AG, isolates, geographical origin, isolation sources were compiled as shown in Appendix A. The primary isolation sources included diseased roots, shoots of soybean crop, and soil (e.g., rhizosphere soil, topsoil) surrounding symptomatic soybean. To create a dataset of all publicly available sequences from the rDNA ITS1-5.8S-ITS2 region associated with *R. solani* AG, we queried National Center for Biotechnology Information (NCBI) GenBank(https://www.ncbi.nlm.nih.gov/genbank; accessed on 12 November 2021) and downloaded all of the sequences for the following studies.

### 2.2. Characterization of the Distribution and Frequency of Anastomosis Groups Assoiated with Soybean

The frequency of AG with known sequences in GenBank from the published literature was calculated using the formula with modification for this study; relative frequency *(F)* = 100 × (*n/N*), in which *n* = the number of each AG/AG subgroup reported in the published literature and *N* = the total number of all AG/AG subgroup reported in the published literature [19,45]. AG showed higher frequency was considered most frequently reported or highly distributed AG or AG subgroup.

### 2.3. Sequences Alignments

The nucleotide sequences of the rDNA ITS1-5.8S-ITS2 region, representing AG were edited and assembled using BioEdit v. 7.2.5 with manual adjustment [46]. The sequences were aligned using the Clustal W algorithm in the Molecular Evolutionary Genetics Analysis (MEGA v. 7.0.26) software, and obvious errors were adjusted manually [47]. The resultant alignments were imported into BioEdit v. 7.2.5 and adjusted manually by visual examination [46]. A sequence of *Atheliarolfsii* FSR-052 (GenBank Accession No. AY684917) (anamorph, *Sclerotiumrolfsii* ) was used as the outgroup [38,39,41,42,43,44,45,46,47,48].

### 2.4. Phylogenetic Analysis

Before conducting phylogenetic analysis, best-fit substitution model selection of the aligned sequences was carried out using the jModelTest v. 2.1.6 package program [49] with model selection strictly based on the Akaike Information Criterion (AIC) estimateand Bayesian Information Criterion (BIC) [50,51]. The Tamura-Nei [52] model was suggested by jModelTest v. 2.1.6. The best-fit substitution model for the phylogenetic trees was mentioned in Appendix A. Phylogenetic trees on the multiple alignments were constructed using MEGA v. 7.0.26. The phylogenetic trees were built using Maximum Likelihood (ML) [53], Neighbor-Joining (NJ) [54] and Maximum-parsimony (MP) [55]. Rates among sites were selected as G (γ distributed) for both ML and NJ. The partial deletion for ML and NJ was set as gap/missing data treatment with a 95% site coverage cut-off, and Nearest-neighbor interchange (NNI) was selected for the heuristic method. The MP analysis was obtained using the Close-Neighbor-Interchange algorithm [55]. Bootstrapping of 1000 random samples from various sequence alignments was used to test each phylogenetic tree’s robustness. Gaps and missing data were removed from all positions. Only nodes with bootstrap values of 70% or higher were shown in the phylogenetic trees. Phylogenetic trees were visualized using the Interactive Tree of Life (iTOLv. 6(http://itol.embl.de/; accessed on 16 November 2021) [56] and Fig Tree v. 1.4.4 (http://tree.bio.ed.ac.uk/software/figtree/; accessed on 17 November 2021) and edited in Adobe Illustrator^®^ CS5 v. 15.0.0 (San Jose, CA, USA).

### 2.5. Principal Coordinate Analysis (PCoA) and Sequence Similarities

Pairwise percentages of sequence similarities of all the isolates within AG and AG subgroups and among the AG and AG subgroups were calculated with the MatGAT v. 2.0 program [57]. Principal coordinate analyses (PCoA) were conducted on pairwise sequences similarity matrix to investigate clustering of AG and AG subgroups using paleontological statistics software package for education and data analysis (PAST v. 4.03) with Gower similarity index [58].

## 3. Results

### 3.1. Distribution of Anastomosis Groups

Nine anastomosis groups (AG) such as AG-1, AG-2, AG-3, AG-4, AG-5, AG-6, AG-7, AG-9 and AG-11, were associated with soybean. According to the geographical distribution, most AG were reported from the United States and Brazil while only a few AG were reported from Japan, India, Canada and Taiwan (Figure 1). Among the AG, AG-1 was the most prevalent and frequently reported AG with a relative frequency of 40%, followed by AG-2 (19.13%), AG-4 (11.30%), AG-7 (10.43%), AG-11 (8.70%), AG-3 (5.22%), AG-5 (3.48%) and AG-6 and AG-9 each with a frequency of 0.87% (Table 1). Similarly, among the AG subgroups associated with soybean, AG-1-IA was frequently reported AG (33.91%), followed by AG-2-2IIIB (12.17%) and each of AG-4-HGII and AG-4-HGIII with a frequency of 4.35%. On soybeans, all of these AG were pathogenic (Table 1). AG-1 caused severe foliar and web blight of soybean, while the rest of the AG were reported to cause damping-off, root, and hypocotyl rot of soybean (Table 1).

### 3.2. Genetic Diversity of Anastomosis Groups

Initially, 115 *R. solani* AG associated with soybean were collected from published literature in this study (Table 1). Only 102 AG with known isolate names, pathogenicity and geographic origins were used to explore genetic diversity and phylogeny (Figure 2, Figure 3 and Figure 4). Using NJ, ML and MP methods, sequence analysis clustered AG into two major clades (Figure 2, Figure 3 and Figure 4). The 102 AG clustered into two distinct clades, with clade I including 62 AG and clade II including 40 AG (Figure 2, Figure 3 and Figure 4). The 62 AG in clade I further clustered into five distinct subclades (Ia, Ib, Ic, Id, and Ie), with sufficient bootstrap support for each subclade (Figure 2, Figure 3 and Figure 4). Clade Ia included isolates of AG-11 (subclade Ia-1) and isolates of AG-5 (subclade Ia-2), clade Ib included isolates of AG-2, clade Ic included isolates of AG-3, clade Id included isolates of AG-4, and clade Ie included isolates of AG-7. Subclades Ia-1 (isolates of AG-11) and Ia-2 (AG-5) within Ia were closely spaced subclades in the tree. Even the subgroups of AG within these subclades clustered separately. For example, subclade clade Ib was further subdivided into three subclades (Ib-1, Ib-2, and Ib-3). Ib-1 included isolates of AG-2-1, Ib-2 included AG-2-2 and Ib-3 included isolates of AG-2-2IIIB. Similarly, clade Id was further subdivided into four subclades, i.e., subclade Id-1 (containing isolates of AG-4-HGI), subclade Id-2 (isolates of AG-4-HGII), subclade Id-3 (AG-4-HGIII), and subclade Id-4 (AG-4-4). The 40 AG in clade II also clustered into two distinct subclades (IIa and IIb). Subclade IIa only included isolates of AG-1-IC and AG-1-IB, while subclade IIb included 38 isolates of AG-1-IA (Figure 2, Figure 3 and Figure 4).

### 3.3. Relationship between Genetic Diversity of Anastomosis Groups and Their Geographic Origin

Besides, closely related AG or AG subgroups associated with soybean were clustered together regardless of the geographical origin from where they had been identified (Figure 2, Figure 3 and Figure 4). For example, isolates of AG-1-IA in subclade IIb from Brazil, Japan, and the USA clustered together (Figure 2, Figure 3 and Figure 4). Similarly, isolates of AG-5 from the USA and Japan, in subclade Ia-2, clustered together. Moreover, isolates of AG-2-2 from the USA and Brazil in the subclade Ib-2 clustered together (Figure 2, Figure 3 and Figure 4). In conclusion, closely related AG associated with soybean were clustered together regardless of the geographical origin from where they had been identified.

### 3.4. Genetic Relatedness among and within Clades and Subclades Representing Anast Mosis Groups

ML, MP and NJ phylogenetic trees showed that AG or AG subgroups were clustered together regardless of the geographical origin. Therefore the percentage of sequence similarities within and among clades and subclades of 108 AG without their corresponding geographical origin, was established by direct pairwise comparisons (Table 2). Within the proposed subclades, all the AG-11 and AG-5 closely related isolates within subclade Ia had the widest range of sequence similarities of 77 to 99.4%, followed by AG-7 closely related isolates within subclade Ie which had sequence similarities of 82.1 to 100%, AG-4 closely related isolates within subclade Id which had sequence similarities of 87.4 to 100% and AG-1 closely related isolates within subclades IIa-1 and IIa-2 which had sequence similarities of 87.5 to 100% (Table 2). All the AG-5 closely related isolates within subclade Ia-2, AG-2 closely related isolates within subclade Ib, and AG-3 closely related isolates within subclade Ic had the sequence similarity greater than 90%. The sequence similarity of AG-11 (subclade Ia-1), AG-5 (subclade Ia-2), AG-2 (subclade Ib), AG-3 (subclade Ic), AG-4 (subclade Id), and AG-7 (subclade Ie) of clade I were higher than isolates of AG-1 of clade II. There were also variations in the percentage of sequence similarities of AG and/or AG subgroups between the subclades. For example, the sequence similarity between subclade Ib-1 (isolates of AG-2-1) and subclade Ib-2 (isolates of AG-2-2) and between subclade Ib-1 (isolates of AG-2-1) and subclade Ib-3 (isolates of AG-2-2IIIB) were 83.7–86% and 81.5–85.4%, respectively. In contrast, the sequence similarity between subclade Ib-2 (isolates of AG-2-2) and subclade Ib-3 (isolates of AG-2-2IIIB) was 92 to 99%. This suggests a closer genetic relatedness between the isolates of AG-2-2 and AG-2-2IIIB than between the isolates of AG-2-1. The AG-4 (Subclade Id) isolates were further sub-clustered into four subgroups (1, HG-I, -II, and -III). The sequence similarity within AG-4 (Subclade Id) as a whole rangesfrom 81.7% to 100%; it is considerably lower than within the subgroups: 99.4, 99.6–100, 87.4–99, and 87.2% for HG-I, -II, -III, and 1 respectively. Furthermore, the percentage of sequence similarity was higher than 87.2% within an AG subgroup (IA, IB, IC, HG-1, HG-II, HG-III, 2-1, 2-2, 2-IIIB), 81.7 to 100% for different subgroups within an AG, and 69.5 to 92.5% among different AG. The above-mentioned phylogenetic trees were further explained and supported by PCoA. PCoA grouped AG into two major clades (Figure 5). Clade I included the isolates of AG-3, 4, 5, 7, and 11, and clade II included isolates of AG-1. Within the clades, subgroups of AG formed a separate cluster. For example, isolates of AG-1-IB and AG-1-IC (subclade IIa) formed distinct clusters than isolates of AG-1-1A (subclade IIb). Similarly, isolates within subclade Id such as AG-4-HGI, AG-4-HGII, and AG-4-HGIII formed separate clusters (Figure 5).

## 4. Discussion

*R. solani* is a soil borne pathogen that affects soybean worldwide and has a significant economic impact in all soybean growing countries [1,7,11,19,21,59,60,61]. All AG associated with soybean reported in this study belonged to *R. solani*. The frequency of AG varied substantially geographically. Most of the AG were reported from the diseased soybean plants in the USA and Brazil. More AG discoveries associated with soybean in the USA may imply an expansion of the host range and genetic diversity of *R. solani* [4]. Furthermore, AG associated with soybean might have been studied more intensively in the USA than other countries because of their greater relative importance as plant pathogen of soybean. In the USA, foliar blight caused by AG-1 and, hypocotyl and root rot caused by AG-2-2IIIB, AG-4, AG-5, AG-3, AG-7, and AG-11 caused as high as 45% soybean yield losses [4,21,61,62,63]. In this study, the reports of a few AG from Japan, Canada, Taiwan, Japan, and India were probably due to a lack of sampling or isolation methods. In Brazil, foliar blight, damping-off, and root rot caused by AG-1 and AG-2 resulted in an estimated 31 to 60% soybean yield loss [8,32,64]. In Canada, root rot ranked fourth among 22 diseases causing severe losses in soybean [65,66]. In India, foliar blight caused by AG-1 caused an average yield loss of 40% to 50% [1,43,67,68]. Besides, AG-2 and AG-5 have been reported to cause hypocotyl rot of soybean in Japan [44]. AG-7 is responsible for the damping-off of soybean seedlings in Taiwan [42]. In recent years, the frequency of legumes in crop rotations has increased, and also the intensive cultivation of soybean might be another reason for increasing the frequency of *R. solani* AG [4,5,21,42,61,62,63].

Besides, our study also revealed the most frequently reported AG from soybean. AG-1 was the most frequently reported AG from soybean, followed by AG-2, AG-4, AG-11, AG-7, AG-3, and AG-5. Frequently reported AG doesn’t indicate whether it is highly pathogenic or not pathogenic on soybean. For example, AG-1-1A is highly pathogenic on soybean in Brazil; however, AG-2-2IIIB, AG-4, and AG-5 are highly pathogenic on soybean in the USA [5,19,20,21,22]. Hence, AG diversity, frequency, and distribution could be influenced by the dynamics of the host-pathogen relationship, genetic flexibility, and degree of adaptation [69]. Furthermore, crop rotation, soil types, soybean cultivars, cropping patterns, and climatic conditions of the particular region may encourage the presence of specific AG over others [70]. In addition, root-associated microbial communities also influence AG distribution [32].

The most reliable approach for phylogenetic analysis and genetic diversity of AG and AG subgroups of *R. solani* is the molecular characterization utilizing the sequences of the rDNA ITS1-5.8S-ITS2 region [5,37,38]. We were able to make conclusions about the phylogenetic relationships among AG and AG subgroups using the sequences of the rDNA ITS1-5.8S-ITS2 region from the NCBI GenBank. In this study, phylogenetic analysis based on MP, NJ and ML showed AG forms two distinct clades. Clade I included isolates of AG-2, AG-3, AG-4, AG-5, AG-7, and AG-11, whereas clade II included isolates of AG-1. Each AG forms a distinct subclade within the clades except AG-5 and AG-11, which form a distinct subclade (Ia).This suggests that isolates of AG-5 and AG-11 may be more closely related to each other. Previous studies have shown that even isolates of AG-5 of soybean clustered with the isolates of AG-11 of other legumes such as lupins [37]. Besides, our study showed that even AG subgroups form distinct subclades. For example, isolates of AG-1C and IB form a sister subclade with isolates of AG-1A within clade IIa (AG-1 isolates). Sequence analysis in previous studies revealed that AG-1-B was genetically distinct from AG-1 IA and IB [69]. Likewise, within the sub clade Ib (AG-2 isolates), isolates of AG-2-1 form a sister subclade with isolates of AG-2-2 and AG-2-IIIB. Previous studies considered AG-2 a polyphyletic with subgroups consistently forming different clades or subclades [71].

AG-2 is a highly heterogeneous AG with substantial genetic diversity and is further divided into nine subgroups such as 1, 2, t, Nt, 2IIIB, 2IV, 2LP, 3, 4 that cause rots and damping-off disease in soybean [37]. Moreover, within subclade Id (AG-4 isolates), few AG-4-HGIII isolates form a sister clade with isolates of AG-4-HGII and AG-4-HGI. Besides, most of the isolates of AG-4-HGII and AG-4-I were clustered together. This indicated that subgroups HGI and HGII were found to be more closely related than subgroup HGIII [15,17,32]. Furthermore, our study also revealed that AG did not have preferences for geographical origin. Most clades or subclades with high bootstrap support indices include AG and AG subgroups from USA, Brazil, and other countries. In a previous study, the authors of the reference [34] analyzed sequences of AG from Europe, North America, Australia and Asia associated with legumes, cereals and vegetables and found that AG did not have a preference for a geographic origin; however, some AG were found to be host-specific [72]. The authors of the reference [73,74,75,76,77,78] showed that isolates of AG from different countries are categorized under the same AG or AG subgroups. Furthermore, the pairwise distance matrix based on sequence similarities revealed that the isolates of AG within the clades and subclades shared high sequence similarities. In contrast, isolates of AG from different clades and subclades showed less similarity. Furthermore, the sequence similarity was higher than 87.2% within an AG subgroup, 81.7 to 100% for different subgroups within an AG, and 69.5 to 92.5% among different AG. These results are consistent with previous studies that assessed the sequence similarities of ITS sequences [15]. They found that sequence similarity was above 96% for the same AG subgroup, 66–100% for different subgroups within an AG, and 55–96% for AG. In addition, PCoA revealed that AG and/or AG subgroups form a separate group from each other. Previous reports showed that the sequence homology in the ITS regions was higher for isolates of the same subgroup than isolates of different subgroups within an AG and isolates of different AG [38]. Our study revealed that the rDNA-ITS sequences were clustered consistently according to their known AG and not according to geographical origin. Cluster analyses based on rDNA-ITS on sequences of *R. solani* AG and AG subgroups associated with a soybean of a specific geographical origin have already been reported [1,7,19,21,61].

## 5. Conclusions, Limitations and Future Directions

In conclusion, this study provides the first documentation regarding the global genetic diversity and distribution of *R. solani* AG associated with soybean. AG-1, AG-2, and AG-4 were the most prevalent and widely documented AG in soybean. AG-1 was responsible for foliar and web blight of soybean, whereas the remaining AG were causing damping-off, root and hypocotyl rot. Across geographical origin, most of the AG were reported from the USA, followed by Brazil. Phylogenetic and genetic diversity analysis revealed that AG and/or AG subgroups formed distinct clades and subclades without corresponding to geographical origin. Pairwise percentages of sequence similarities within AG and subgroups and principal coordinate analysis also support the phylogenetic and genetic diversity analysis. The rDNA ITS1-5.8S-ITS2 region has been successfully sequenced and phylogenetically analyzed to reliably separate *R. solani* isolates into several groups and subgroups that correspond to the various AG [5,15,17,18,35]. However, sequence analysis of the rDNA ITS1-5.8S-ITS2 region is not without its attendant limitations. Though the differences in the rDNA ITS1-5.8S-ITS2region are sufficiently large to differentiate the AG reliably, they could not differentiate isolates of the same AG [75]. Furthermore, researchers do not verify or validate sequences deposited in databases and repositories; depositing an incorrectly named AG is almost inevitable. Complete information about isolate name, host, or geographical origin may not be included. Besides, the rDNA ITS1-5.8S-ITS2 region would not always be ideal because of high mutation rates. Furthermore, *R. solani* is multinucleate; therefore, there is the possibility of numerous nucleotide variations in this region even in the single strain of *R. solani* [76,77,78]. Hence, the genetic diversity and phylogeny of AG must be augmented with additional sequences such as large-subunit rRNA (LSU) region, ß-tubulin, the largest (*RPB1*) and the second-largest (*RPB2*) subunits of RNA polymerase, translation elongation factor (*tef-1α*), the mini-chromosome maintenance protein (*mcm7*), calmodulin (*CaM*), and topoisomerase I (*top1*) gene. Furthermore, studies involving genomic, transcriptomic, proteomic and mitogenomic analysis may provide insights into the phylogeny and genetic diversity of *R. solani* AG.

## Figures and Tables

**Figure 1 genes-13-02417-f001:**
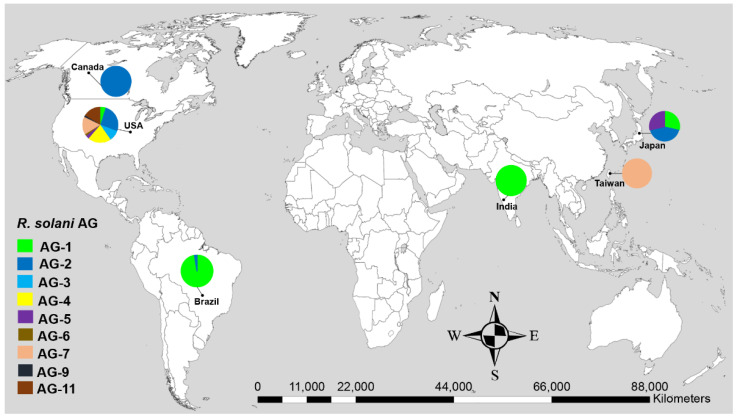
Geographical distribution of *R. solani* anastomosis groups (AG) associated with soybean with known sequences in GenBank. Data in pie charts indicate the number of AG reported in each geographical origin.

**Figure 2 genes-13-02417-f002:**
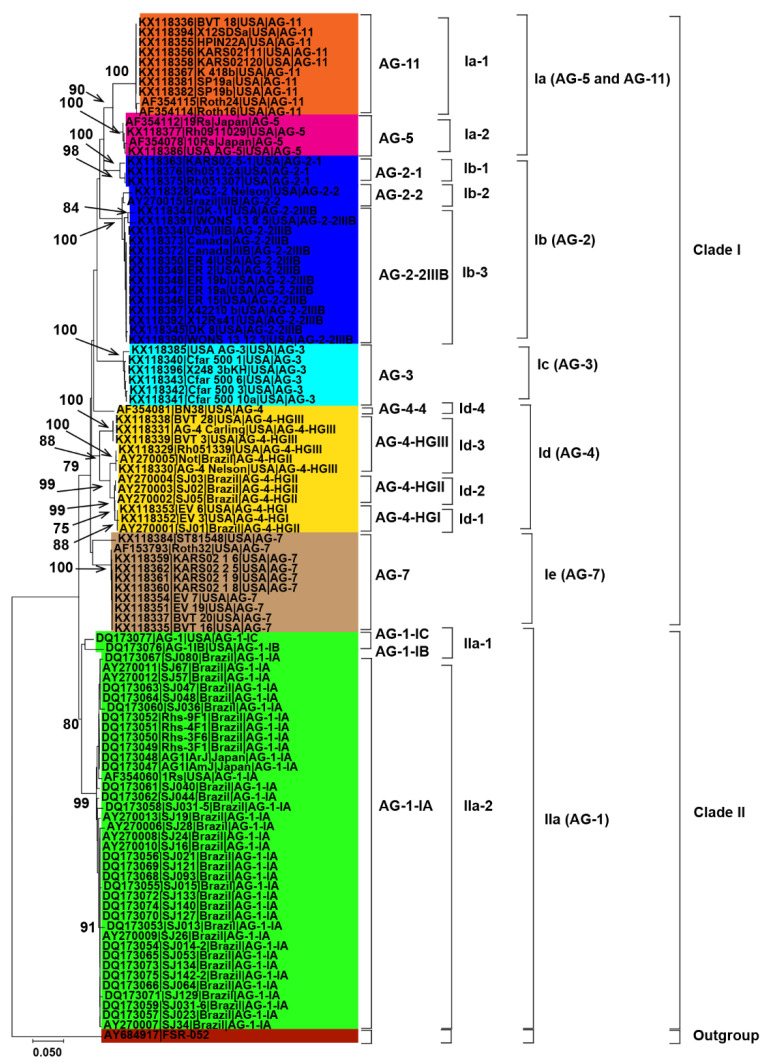
Genetic relatedness among the AG from soybean. Neighbor-Joining (NJ) analysis was used to build the trees. The accession numbers are followed by isolate, geographical origin and AG. Different colors show clades and/or subclades associated with AG. Thousands of replications were used to bootstrap tree branches. Numbers at nodes indicate bootstrap values, and only those ≥70 are shown. The outgroup, *Athelia rolfsii* FSR-052 (GenBank Accession No. AY684917), was used to root the tree. Scale bar indicates a genetic distance of 0.05 for horizontal branch length.

**Figure 3 genes-13-02417-f003:**
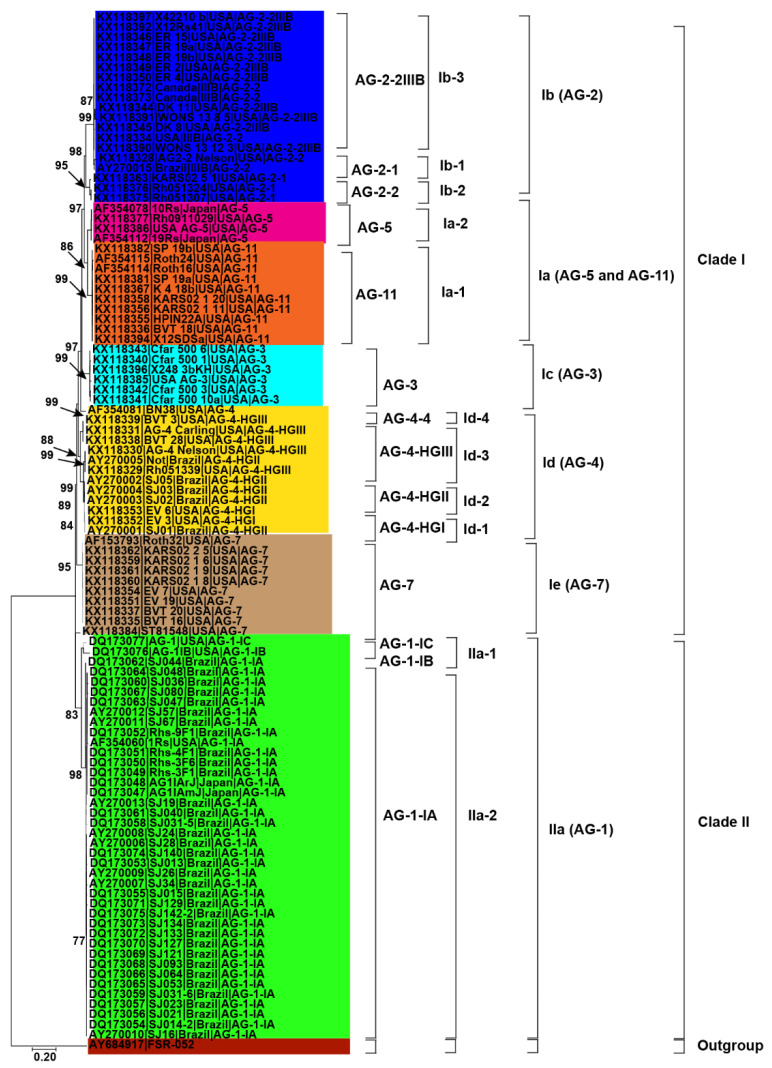
Genetic relatedness among the AG from soybean. Maximum likelihood (ML) analysis was used to build the trees. The accession numbers are followed by isolate, geographical origin and AG. Different colors show clades and/or subclades associated with AG. Thousands of replications were used to bootstrap tree branches. Numbers at nodes indicate bootstrap values, and only those ≥70 are shown. The outgroup, *Athelia rolfsii* FSR-052 (GenBank Accession No. AY684917), was used to root the tree. Scale bar indicates a genetic distance of 0.2 for horizontal branch length.

**Figure 4 genes-13-02417-f004:**
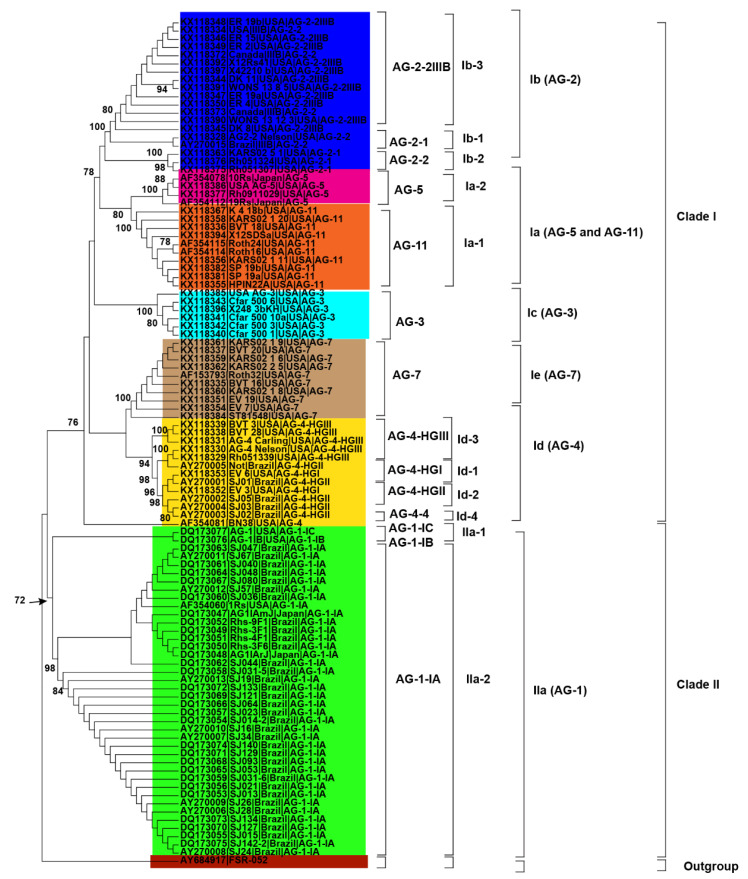
Genetic relatedness among the AG from soybean. Maximum parsimony (MP) analysis was used to build the trees. The accession numbers are followed by isolate, geographical origin and AG. Different colors show clades and/or subclades associated with AG. 1000 replications were used to bootstrap tree branches. Numbers at nodes indicate bootstrap values, and only those ≥70 are shown. The outgroup, *Athelia rolfsii* FSR-052 (GenBank Accession No. AY684917), was used to root the tree.

**Figure 5 genes-13-02417-f005:**
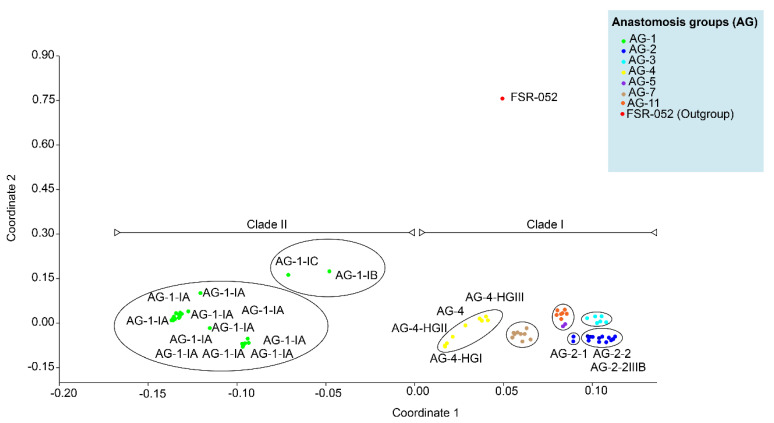
Principal coordinate analyses (PCoA) ordination of *R.solani* anastomosis groups. (AG) based on a Gower distance of sequence similarity matrix.

**Table 1 genes-13-02417-t001:** Frequency of *R. solani* AG associated with soybean across the geographical origins.

AG	AG Subgroups	Geographical Origin								
		USA	Brazil	Canada	Taiwan	India	Japan	Total	^a^ *F*	

**AG-1**	AG-1-IA	1	36				2	39	33.91	40.00
	AG-1-IB	1				2		3	2.61	
	AG-1-IC	1						1	0.87	
	AG-1-IF		1					1	0.87	
	1					1		1	0.87	
	AG-1-ID		1					1	0.87	
**AG-2**	AG-2-1	3						3	2.61	19.13
	AG-2-2	1	1					2	1.74	
	AG-2-3						3	3	2.61	
	AG-2-2IIIB	12		2				14	12.17	
**AG-3**		6						6	5.22	5.22
**AG-4**	AG-4	1						1	0.87	11.30
	AG-4-HGI	2						2	1.74	
	AG-4-HGII	5						5	4.35	
	AG-4-HGIII	5						5	4.35	
**AG-5**		2					2	4	3.48	3.48
**AG-6**		1						1	0.87	0.87
**AG-7**		10			2			12	10.43	10.43
**AG-9**		1						1	0.87	0.87
**AG-11**		10						10	8.70	8.70
**Total**		62	39	2	2	3	7			

^a^*F* (Relative frequency) = 100 × (*n* /*N* ), in which *n* = the number of each AG/AG subgroup and *N* = the total number of all AG/AG subgroups.

**Table 2 genes-13-02417-t002:** Percentages of sequence similarities of rDNA-ITS sequences of *R. solani* AG within and between clades and subclades.

Clades and Subclades		Ia (AG-11 and AG-5)		Ib (AG-2)			Ic (AG-3)	Id (AG-4)				Ie (AG-7)	IIa (AG-1)		Outgroup
		Ia-1	Ia-2	Ib-1	Ib-2	Ib-3		Id-1	Id-2	Id-3	Id-4		IIa-1	IIa-2	
**Ia (AG-11 and AG-5)**	**Ia-1**	77–99.6													
	**Ia-2**	85.8–93	93–99.4												
**Ib (AG-2)**	**Ic-1**	84.6–89.18	81.9–89	97.2–99.4											
	**Ib-2**	80.7–85.1	76.1–84.6	83.7–86	90.9–96.5										
	**Ib-3**	79.4–85.7	74–85.1	81.5–85.4	92–99.1	91.6–100									
**Ic (AG-3)**		82.3–88.3	83.8–84.9	91.4–92.5	82.7–85.2	79.3–85.7	97.7–99.4								
**Id (AG-4)**	**Ie-1**	80.7–84.6	80.1–80.2	85.1–85.9	82–82.8	77.2–83.8	81.8–84.9	99.4 *							
	**Id-2**	79.3–80.8	79.4–84.2	80.7–81.9	78.3–80.5	75.8–82.5	78.3–80.6	91.1–94	99.6–100						
	**Id-3**	79.5–84.1	76–84.5	82.5–86	74.6–83.3	76.2–84.3	81.7–83.89	87.1–96.4	81.7–94.4	87.4–99.1					
	**Id-4**	79.1–82.8	79.6–83.6	81.8–83.3	75.5–77.1	72.5–77.1	81.9–82.2	82.6–82.9	86.1–86.4	82.6–83.6	87.2 *				
**Ie (AG-7)**		77.2–83.8	76–84.6	83.3–86.3	79.9–85.9	74.7–82.3	78.4–85.9	81.5–87.9	80–83.7	84.4–87.5	83.1–84.6	82.1–99.1			
**IIa (AG-1)**	**IIa-1**	72.5–79.8	74–80.5	74.7–77.8	69.6–75.2	69.5–77.5	73.3–76.5	76.2–77.9	81.7–83.4	74.4–78.5	79.4–81.6	74.3–85.5	87.9–100		
	**IIa-2**	72.8–79.4	73–83.7	75.8–80.6	69.5–80.5	70.4–80.2	73–79.8	77.9–82.9	81.9–88.8	75.7–83.3	81.7–82.8	74.7–83.6	76.3–92.2	89.5–100	
**Outgroup**		47.4–51.3	46.2–49.1	50–51	47.3–52.7	48.4–52.5	51.5–52.5	51.5–51.9	46.6–48	51.1–53.1	48.3 *	48.8–49.3	39.1–42.3	40.9–46	100 *

* Range could not be calculated for one isolate having one sequence.

## Data Availability

Data sets analyzed during the current study are available from the corresponding author on reasonable request.

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
