# Peer review of "Population Genetics and Anastomosis Group’s Geographical Distribution of Rhizoctonia solani Associated with Soybean"

_genes, 2022, doi:10.3390/genes13122417_

Round 1

Reviewer 1 Report

General evaluation of the publication

1. It would be appropriate to shorten the title of the article to “Population genetics and anastomosis group’s geographical distribution of Rhizoctonia solani associated with soybean”

2. This part “Distribution of anastomosis groups in The World” It is not included in studies in some countries such as Turkey. Perhaps the reasons for this should be stated in the discussion section.

3. Some corrections need to be made in the text. For example, fungi species should be written in italics.

Note: Some corrections have been made on the publication.

Author Response

Point By Point Response to Reviewer-I Comments and Suggestion

  1. It would be appropriate to shorten the title of the article to “Population genetics and anastomosis group’s geographical distribution of Rhizoctonia solani associated with soybean”

Response: Thank you very much for your valuable suggestion, the title of the article has been amended and edited as per the light of suggestion. Now the new title of the article is “Population genetics and anastomosis group’s geographical distribution of Rhizoctonia solani associated with soybean”. Thanks

  1. This part “Distribution of anastomosis groups in The World” It is not included in studies in some countries such as Turkey. Perhaps the reasons for this should be stated in the discussion section.

Response: Thank you for your valuable comment, the suggested part of the “Distribution of anastomosis groups in The World” has been added up as per available information in the discussion section. Nine anastomosis groups (AG) such as AG-1, AG-2, AG-3, AG-4, AG-5, AG-6, AG-7, AG-9 and AG-11, were associated with soybean. According to the geographicaldistribution, most AG were reported from the United States and Brazil. While only a few AG were reported from Japan, India, Canada and Taiwan (Figure 1). Among the AG, AG-1 was the most prevalent and frequently reported AG with a relative frequency of 40%, followed by AG-2 (19.13%), AG-4 (11.30%), AG-7 (10.43%), AG-11 (8.70%), AG-3 (5.22%), AG-5 (3.48%) and AG-6 and AG-9 each with a frequency of 0.87% (Table 1). Similarly, among the AG subgroups associated with soybean, AG-1-IA was frequently reported AG (33.91%), followed by AG-2-2IIIB (12.17%) and each of AG-4-HGII and AG-4-HGIII with a frequency of 4.35%. On soybeans, all of these AG were pathogenic (Table S1). AG-1 caused severe foliar and web blight of soybean, while the rest of the AG were reported to cause damping-off, root, and hypocotyl rot of soybean (Table S1).

  1. Some corrections need to be made in the text. For example, fungi species should be written in italics.

Response: Thank you for your valuable comment, all the text of the article has been gone through again critically and the mistakes related to species name non italic has been made italic in the complete manuscript to improve its quality. Thanks.

4- Note: Some corrections have been made on the publication.

Response: Thank you very much, the corrections has been acknowledged and again rectified through track changes and English language quality was improved by the native English speaker.

Reviewer 2 Report

The present manuscript entitled ‘Population genetics and anastomosis group’s geographical distribution of soil and seed borne fungus Rhizoctonia solani associated with soybean’ is written well. The manuscript is based on In silico analysis of sequences so, title of the manuscript should reflect that it is an In silico analysis.

Apart from this some minor corrections are needed to improve the manuscript as mentioned below:

1.      Line : 99- ‘soybean mentioned above, we aimed to’ (to should be deleted)

2.      Line: 189-Initially, 115 R. solani AG associated with soybean were collected from (R. solani should be in italic)

Author Response

Point By Point Response to Reviewer-II Comments and Suggestion

The present manuscript entitled ‘Population genetics and anastomosis group’s geographical distribution of soil and seed borne fungus Rhizoctonia solani associated with soybean’ is written well. The manuscript is based on In silico analysis of sequences so, title of the manuscript should reflect that it is an In silico analysis.

Response: Thank you very much for your encouraging remarks regarding the article, The title of the article has been modified keeping in view the suggestions provided, and it is “Population genetics and anastomosis group’s geographical distribution of Rhizoctonia solani associated with soybean”. Thanks

Apart from this some minor corrections are needed to improve the manuscript as mentioned below:

Response: Thanks much for highlighting improvements, the complete manuscript was gone though critically and all the mistakes were rectified to improve the quality of the article.  The English language editing was also performed by a native English speaker.

  1. Line : 99- ‘soybean mentioned above, we aimed to’ (to should be deleted)

      Response: Thank you for your valuable comment, the correction has been made and now this seems “Considering the genetic diversity and different diseases causing abilities of R. solani AG on soybean mentioned above, the current study was aimed 1) To determine most frequently reported and dominant AG associated with soybean; 2) To explore the genetic diversity of AG based on rDNA ITS1-5.8S-ITS2 sequence analysis; 3) To determine the relationship between geographical origin and genetic diversity of AG.

  1. Line: 189-Initially, 115 R. solani AG associated with soybean were collected from (R. solani should be in italic)

      Response: Thank you for your valuable comment, the suggested correction has been amended and rectified not only at this place but also wherever it was found appropriate. Thanks. 
